# Theoretical Study on Improving the Catalytic Activity of a Tungsten Carbide Surface for Hydrogen Evolution by Nonmetallic Doping

Ye Su [1,†], Xilin Xiong [2,†], Xiaoxu Wang [1] , Keke Song [1], Yating Zhou [1], Qing Liu [1] and Ping Qian [1,*]

[1] Beijing Advanced Innovation Center for Materials Genome Engineering, Beijing Key Laboratory for Magneto-Photoelectrical Composite and Interfere Science, School of Mathematics and Physics, University of Science and Technology Beijing, Beijing 100083, China; yjsu@ustb.edu.cn (Y.S.); wangxx@bcc.ac.cn (X.W.); KeKSong@163.com (K.S.); suyechina@163.com (Y.Z.); liuqing6903@163.com (Q.L.)

[2] Corrsion and Protection Center, University of Science and Technology Beijing, Beijing 100083, China; xlxiong@ustb.edu.cn

* Correspondence: pingqian@ustb.edu.cn

† Both authors contributed equally to this work.

**Abstract:** Tungsten carbide (WC) has received widespread attention as a new type of nonprecious metal catalyst for hydrogen evolution reaction (HER). However, it is still a challenge to improve the surface HER catalytic activity. In this work, the effects of different nonmetallic dopants on the catalytic activity and stabilities of WC (0001) surface for HER were studied by first principles methods. The effects of different types of non-metal (NM = B; N; O; P and S) and doping concentrations ($n_i$ = 25–100%) on HER catalytic activity and stability were investigated by calculating the Gibbs free energy of hydrogen adsorption ($\Delta G_H$) and substitution energy. It was found that the catalytic performance can be improved by doping O and P non-metallic elements. Especially, the $\Delta G_H$ with P doped is −0.04eV better than Pt (−0.085 eV), which is a potential ideal catalyst for HER. Furthermore, the electronic structure analysis was used to explore the origin of the regulation of doping on stability and catalytic activity. The results show that nonmetallic doping is an effective strategy to control the catalytic activity, which provides theoretical support for the future research of HER catalysts.

**Keywords:** nonmetallic doping; WC (0001) surface; HER; DFT

## 1. Introduction

As energy consumption increases, the excessive exploitation of fossil fuels and the release of $CO_2$ after combustion has led to the destruction of the surface environment, in which the greenhouse effect continues to increase. Given this challenge, the development of efficient and clean new energy is an effective solution for the above. Hydrogen energy has drawn significant attention in recent years due to its environmental friendliness, relatively high energy density, and high utility value in modern society. Water electrolysis is a convenient method for producing hydrogen energy [1,2], which provides a channel for the reduction of water to hydrogen molecules as a sustainable energy supply. However, efficient and stabilized catalysts are required to achieve a sufficient reaction rate. Currently, platinum (Pt) is the most effective catalyst [3]. However, due to its high price and the strong adsorption of CO on surface active sites of Pt, which leads to its poisoning, the wide use of Pt catalysts is difficult. Therefore, it is especially important to research and develop high-efficiency electrocatalysts with low overall cost and high resistance to poisoning.

Nonprecious materials that have been explored as electrocatalysts for this reaction include transition metal sulfides [4], phosphides [5,6], and oxides [7]. Among studied nonprecious metal catalysts, the dissociative adsorption of water on the NiO (111) surface is thermodynamically favorable and fast, according to density functional theory (DFT) theoretical calculation [8]. However, this material strategy poses limitations [9]. Among them, transition metal carbides (TMCs) are attracted much attention due to their high conductivity, metallic band states, tunable surface/bulk architectures [10–12]. There have been many previous studies on TMC [13,14]. Levy and Boudart [15] first reported the catalytic activity of tungsten carbide (WC) in the 1970s, and they found that the surface electronic properties of WC were modified by carbon in such a way that WC resembled those of Pt. Dong [16] pointed out that WC had a high ratio of chemical bond energy to surface carbon and thus had similar surface and catalytic properties as those of noble metals.

Recent rising interest has been in the support of a metal overlayer on metal carbides for improving catalytic activity. TMCs loaded with trace amounts of noble metals to produce composite electrocatalytic materials, such as Pt/WC, Au/WC, and Ni/TiC, have been increasingly studied by researchers [17]. As science and technology have developed, it has been found that doping can also improve the catalytic activity of materials. Haiyan Jin et al. [18] showed that transition metal (Co, Ni, and Fe)-doped WC exhibited improved catalytic activity. Metal-doped catalysts, although having the advantages of high catalytic activity, have disadvantages, such as poor stability and low selectivity, which hinder the further application of such materials. Nonmetal doped catalysts avoid such disadvantages and exhibit very substantial performance in overall water decomposition. Robert B. Wexler et al. [19] proved that doping the surface of $Ni_2P$ (0001) with nine nonmetallic elements (B, C, N, O, and so on) could effectively enhance the catalytic activity of the $Ni_2P$ (0001) surface. Böhm indicated that hydrogen adsorption was activated on partially oxidized WC electrodes after anodic polarization [20]. However, few studies have systemically studied nonmetallic element doping in WC.

In this paper, the WC (0001) surface structure was constructed. Then, the surface stability and catalytic activity of the hydrogen evolution reaction were studied by substituting the C on the WC (0001) surface with nonmetallic elements (B, N, O, P, and S). Finally, the origin of the regulation of stability and catalytic activity by doping was studied from the perspective of the electronic structure analysis of the solid surface.

## 2. Results and Discussion

### 2.1. Stoichiometric Tungsten Carbide Single-Crystal Surface

In the actual calculation, the single layer and double layer are constructed on the surface of WC to test the number of layers. The number of odd layers is 3–15, and that of even layers is 4–16. When the even layer model reaches four layers, the energy of WC (0001) tends to be flat, which indicates that it has the properties of bulk material. When the odd number layer model reaches more than five layers, the surface energy of WC (0001) tends to be flat, which indicates that it has the properties of bulk material. In this study, the six layers model on WC (0001) surface is selected for discussion, as shown in reference [21].

Due to breaking symmetry, the surface atoms will undergo stress-strain, which leads to the rearrangement of surface atoms. A large number of calculations and experimental results show that only the top-most surface atoms are clearly rearranged. Therefore, we focus on the structural relaxation between the top three layers of the WC (0001) structure during the configuration optimization process. The variation in the distance between the layers *i* and *j* before and after relaxation $\Delta d$ is defined as follows:

$$\Delta d = ((d'_{ij} - d^0_{ij})/d^0_{ij}) \times 100\% \tag{1}$$

where $d'_{ij}$ is the layer distance between the adjacent *i*-th and *j*-th layers after relaxation and $d^0_{ij}$ is the layer distance between the adjacent layers of the ideal crystal before relaxation [22]. It can be seen from Table 1 that the C-terminated WC (0001) change is obvious. For the WC (0001) C-termination, the

structural relaxation between the first two layers is 21.7%, while between the second and third layers it is 6.23%.

**Table 1.** Calculated percent of relaxation of the $\Delta d_{12}$ (distance between the first two layers) and the $\Delta d_{23}$ (distance between the second and third layers) for the W- and C-terminated WC (0001) surface, respectively, and the literature data presented for comparison.

| Surface Termination | | Δd12/% | Δd23/% | Reference |
|---|---|---|---|---|
| W | This work | −5.11 | 1.68 | - |
| - | Literature data | −4.33 | 2.88 | [17] |
| | - | −4.09 | 1.97 | [23] |
| | - | −2.81 | 2.11 | [24] |
| C | This work | −21.7 | 6.23 | - |
| - | literature | −22.5 | 4.54 | [17] |
| | - | −22.82 | 4.31 | [23] |
| | - | −18.8 | 4.92 | [24] |

It can be seen from the above data that the uppermost two layers of the C-terminated WC (0001) surface are more relaxed, and the relaxation of the second and third layers decreases; so, in the next calculation, we only need to relax the uppermost two layers of atoms. In addition, our calculation results are similar to those of ref. [23].

## 2.2. Effect of Doping on the Surface Stability and H-Adsorption Strength

Stabilization is an important parameter for evaluating catalyst performance. Therefore, we evaluate the stability of different doping configurations. To do this, we summarize the substitution energies ($\Delta E_{sub(i), i = 0.25, 0.50, 0.75, 1.00}$) at different doping concentrations (Step2 of Figure 1), which can be expressed as surface stability [21]. A negative value of $\Delta E_{sub(i)}$ indicates that the system becomes more stable after nonmetal doping, and a positive value indicates that the system becomes unstable after nonmetal doping. Figure 2a shows that $\Delta E_{sub(i)}$ strongly depends on the doping concentration ($n_i$), and we can observe two trends: (1) the O-doped surface becomes more stable, and (2) the B-, N-, P-, and S-doped surface becomes unstable. In addition to nonmetal O, the stability of the surface $C_{1-n_i}X_{n_i}/WC(0001)$ structure can be increased sequentially with $\Delta E_{sub(i)}$ ($i = 0.25$ ML(monolayer)) $< \Delta E_{sub(i)}$ ($i = 0.50$ ML) $< \Delta E_{sub(i)}$ ($i = 0.75$ ML) $< \Delta E_{sub(i)}$ ($i = 1.00$ ML). For the other four unstable nonmetallic doping structures, the $\Delta E_{sub(i)}$ of N element doping structure is relatively small, and thus is relatively stable.

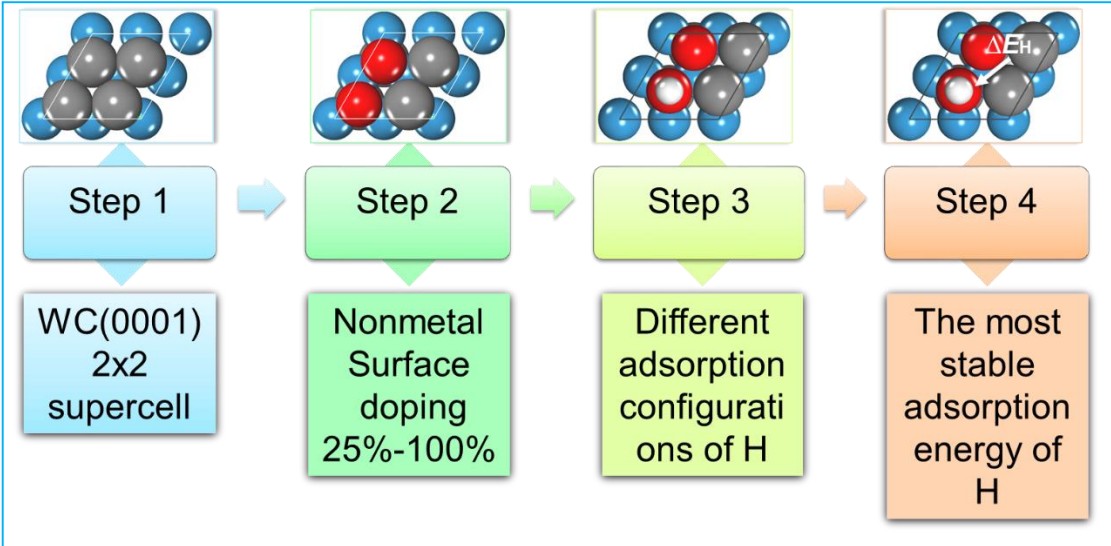

**Figure 1.** The model workflow of this work.

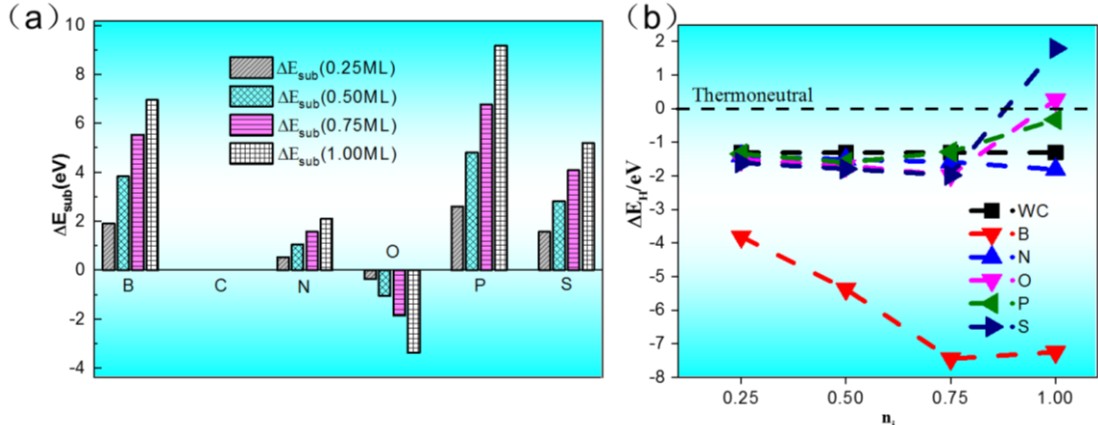

**Figure 2.** (**a**) Dopant substitution energy ($\Delta E_{sub(i),\, i\, =\, 0.25,\, 0.50,\, 0.75,\, 1.00}$) for different nonmetallic doped elements and concentrations and (**b**) The adsorption energy of H ($\Delta E_H$) on the nonmetallic element (B, C, N, O, P and S)-doped WC surface as a function of the surface doping concentration.

Next, in the HER study, we considered H-stabilized adsorption sites with various doped elements and concentrations (Step3 of Figure 1). There are six positions, including the top-site, hollow-site, bridge-site, anti-bridge-site, hex-site, and anti-hex-site, as shown in Figure 6b. The most stable adsorption site of H atom for different nonmetal doped elements and doping concentrations and the corresponding H adsorption energy ($\Delta E_H$) are shown in Table 2 (Step4 of Figure 1). It can be seen that for non-metal N, O, P, and S doping structure, the most stable adsorption sites are all top site. The adsorption site of nonmetallic B is unique. When $n_i$ is 75% and 50%, the optimal adsorption position is bridge-site. When $n_i$ is 25% and 100%, the optimal adsorption position is anti-hex. Besides, we can see that the absorption energy of B doping structure is so favorable. The explanation is as follows: as seen in Figure 2a, the $E_{sub}$ of B doping structure is a more positive value, which indicates that the system becomes less stable after nonmetal doping. Thus, the B doping structure is slightly distorted, which leads to its adsorption configuration and adsorption energy different from other doping structures. Besides, the valence electron number of B element is less than that of the C element, while the valence electrons of other doped elements (N, O, P, and S) are more than the C element. The above results show that B doping is favorable. Furthermore, as doping concentration increases, the doping structure becomes more unstable.

Furthermore, we studied the effect of different dopants and concentrations on the H adsorption energy ($\Delta E_H$) at stable adsorption sites, as shown in Figure 2b. We observe that the three trends are (1) B, (2) N, (3) O, P, and S. The first group (B) $\Delta E_H$ exhibits sharp changes with alterations in concentration; $\Delta E_H$ decreases with increasing concentration $n_i$ from 25% to 75%, while $\Delta E_H$ starts to increase at $n_i = 100\%$. Conversely, there is no significant difference between the N-doped and undoped surfaces (the undoped WC surface is marked with a black square in the Figure 2b). In the third group (O, P, and S), at $n_i = 25$–75%, $\Delta E_H$ shows a small decrease and then significantly increases at $n_i = 100\%$. The increase in S is more obvious than the trends of O and P. At $n_i = 100\%$, $\Delta E_H$ is 1.7918 eV (S), 0.2596 eV (O), and −0.32745 eV (P), respectively.

To further study the source of the outstanding activity of the concentration of nonmetal doping at the atomic level, the Gibbs free energy of hydrogen adsorption ($\Delta G_H$) was calculated. Theoretically, the HER path can be described as containing the initial state H + e⁻, the intermediate state of adsorption H * (* is the adsorption site), and the final state of the 1/2 H₂ product [25,26]. In general, a catalytic performance of $\Delta G_H$ that is close to zero is ideal [27]. It can be seen from our results that compared with WC itself, replacing C with P and O dopants improves the catalytic activity of HER ($\Delta G_H$ is closer to zero). As shown in Figure 3a, the Gibbs free energy of the hydrogen adsorption ($\Delta G_H$) of WC(0001), WC(0001)_O4, WC(0001)_P4, and Pt are −0.71 [28], 0.55, −0.13, and 0.09 eV [29,30], respectively. Besides, the results show that the nonmetal-doped WC structures have good HER catalytic activity when the

complete terminating carbon layer is replaced e.g., by oxygen. Besides, recent rising interest has been in the support of a metal overlayer on metal carbides for improving catalytic activity [16,17]. Thus, it can also be called an effect of overlayer coating.

**Table 2.** The most stable adsorption site of H atom for different nonmetal doped elements and doping concentrations and the corresponding H adsorption energy ($\Delta E_H$).

| Dopant | Doping Density | Structures | Adsorption Energy/eV |
|--------|---------------|------------|----------------------|
| B | 25% | anti-hex | −3.81425 |
|   | 50% | bridge | −5.37456 |
|   | 75% | bridge | −7.45212 |
|   | 100% | anti-hex | −7.25006 |
| N | 25% | top | −1.41112 |
|   | 50% | top | −1.51332 |
|   | 75% | top | −1.58855 |
|   | 100% | top | −1.81286 |
| O | 25% | top | −1.51835 |
| - | 50% | top | −1.69721 |
|   | 75% | top | −1.98577 |
|   | 100% | top | 0.25959 |
| P | 25% | top | −1.34954 |
|   | 50% | top | −1.60193 |
|   | 75% | top | −1.28851 |
|   | 100% | top | −0.32748 |
| S | 25% | top | −1.61477 |
|   | 50% | top | −1.79602 |
|   | 75% | top | −1.98137 |
|   | 100% | anti-hex | 1.79181 |
| WC | 0.00 | top | −1.30734 |

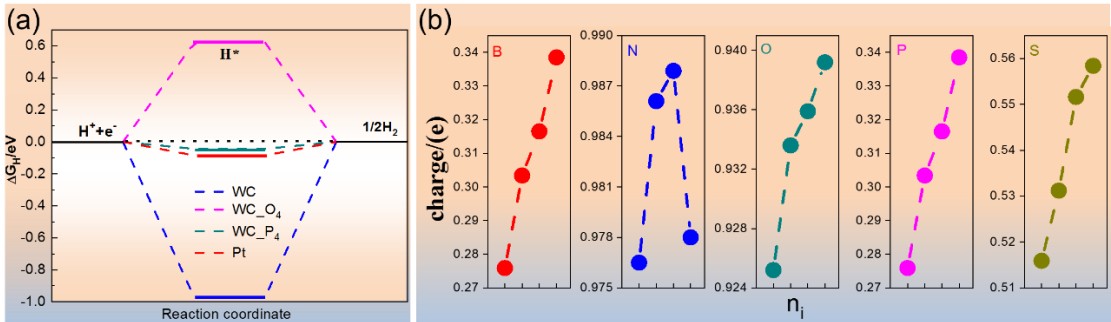

**Figure 3.** (**a**) Gibbs free energy of the hydrogen adsorption ($\Delta G_H$) profile of the HER on different catalysts and the (**b**) effect of various dopants and their surface concentrations on the average nonmetal charge.

To better understand the effect of the nonmetal doping concentration ($n_i$) on $\Delta G_H$ and $\Delta E_{sub(i)}$, we used charge transfer as a descriptor for research and analysis. In Figure 3b, the charge means the charge change of nonmetallic element after doping. Besides, N and O doped systems have the highest charge change since N and O elements have the highest electronegativity and the strongest electron acquisition capability among doping elements. Figure 3b shows approximately linear relationships between the charge transfer and doping concentration ($n_i$), and the slopes can be interpreted as the direction of electron transfer between W and the doped nonmetal elements [16]. For example, a positive slope corresponds to the electron transfer from W to the doped nonmetal, oxidized W (electron donor) and reduced doped nonmetal (electron acceptor). Conversely, if the slope is negative, the electron transfer goes the opposite way as above. However, the electron transfer and $\Delta G_H$ are not strongly correlated. For example, when doping nonmetallic O, the tendency of electron transfer and the trend

of $\Delta G_H$ are the same. However, when doping nonmetallic B, a good tendency for electron transfer is observed, but the trend of $\Delta G_H$ is not uniform. Therefore, it is not appropriate to use the trend of electron transfer to explain changes in $\Delta G_H$.

Single charge transfer cannot describe the regulation effect of nonmetallic element doping on hydrogen evolution, which may be because doping not only regulates the surface electronic structure, but also produces surface local structure distortion, as shown in Figure 4. From the analysis of electron localization function (Figure 4a–c), it is found that WC (0001), WC (0001)_O4, and WC (0001)_P4 are obviously different, which means that the bonding strength of surface nonmetallic elements is different. From Figure 4e–f, it is found that from WC (0001), WC (0001)_O4 to WC (0001)_P4, the bond length between the surface nonmetal and WC base gradually increases, and the electron density distribution is also different. The change of surface crystal structure and electronic structure caused by this doping explains why the O-doped system becomes more stable, while the P-doped system becomes unstable. The synergistic effect caused by the doping band further regulates the catalytic activity of hydrogen evolution on WC surface.

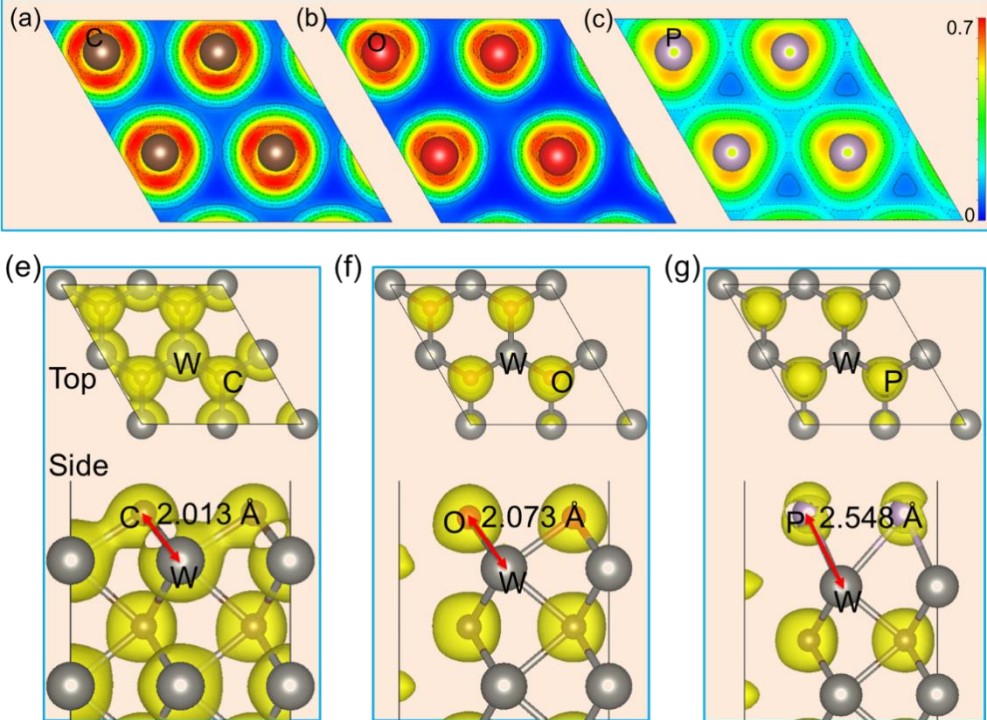

**Figure 4.** ELF and crystal structure and charge density for (**a**,**e**) WC (0001), (**b**,**f**) WC (0001) _O4, and (**c**,**g**) WC (0001) _P4, respectively.

To analyze the origin of the change in the HER catalytic activity, the density of state (DOS) was obtained by analyzing the electronic structures of WC(0001), WC(0001)_O4, and WC(0001)_P4. The total density of state (TDOS) values and projected density of state (PDOS) values of the above structures are shown in Figure 5a–c. As shown in Figure 5c, significant changes occur near the Fermi level. When the doping concentration of nonmetal O is 4, a significant peak appears near −1 eV, and when the doping concentration of nonmetal P is 4, a characteristic peak at −0.3 to the Fermi level (0 eV) forms. Thus, W-5d and P-2p undergo orbital hybridization, and the formation of characteristic peaks indicates that there is a significant charge transfer between the W and P atoms; this result further indicates that the WC(0001)_P4 $\Delta G_H$ value is close to that of Pt because the charge transfer occurs similarly based on the electronic structure.

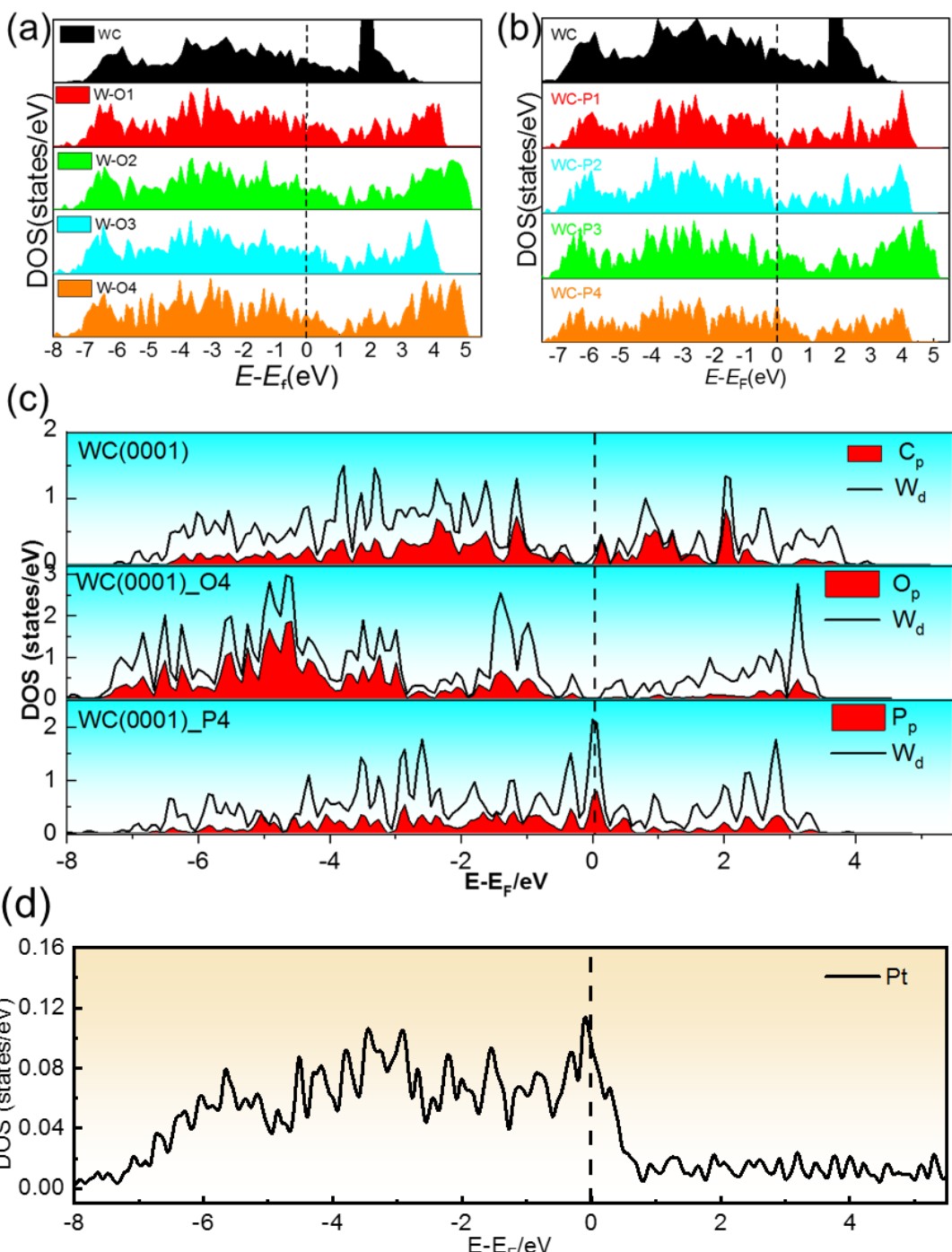

**Figure 5.** (**a**) Calculated total DOS for WC(0001) and WC(0001)_O1,2,3 and 4; (**b**) WC(0001)_P1,2,3 and 4; (**c**) PDOS of WC(0001), WC(0001)_O4 and WC(0001)_P4 for the C, P, and O p-states and W d-states; (**d**) Calculated DOS for Pt.

The DOS diagram analysis of the total density of state values of nonmetallic O and P are compared with the Pt and undoped WC (0001) surfaces. The result shows that the DOS pattern of WC (0001)_P4 is near the Fermi level, and it is most similar to Pt because the characteristic peak also significantly increases compared to that of the WC (0001) surface (Figure 6c,d). This study shows that the catalytic activity of replacing C with nonmetallic P is good, and the $\Delta G_H$ of WC(0001)_P4 is the closest to Pt [31], thereby exhibiting good HER catalytic activity. The above analysis of $\Delta G_H$ and electronic structures shows that nonmetallic doping is an effective strategy to improve catalytic activity.

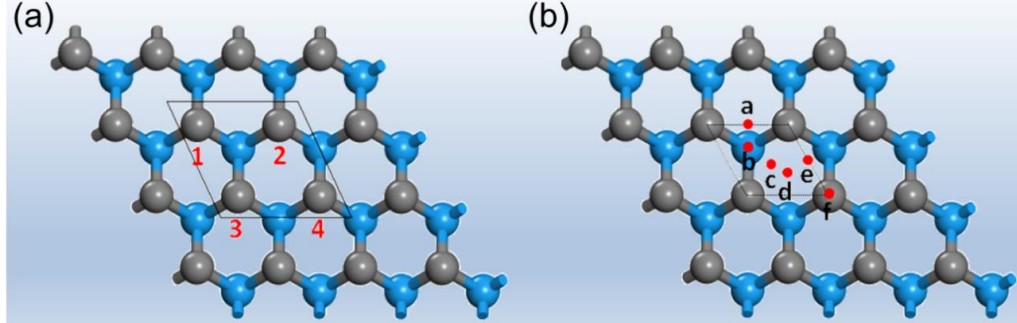

**Figure 6.** The (2 × 2) supercell structure of the C-terminated WC(0001) surface. (**a**) The number of the labeled C atoms indicates the preferred sequence of substitution with dopants. (**b**) The six sites that bind H. a, b, c, d, e and f, which are named the bridge-site, hex-site, hollow-site, anti-hex-site, anti-bridge-site and top-site, respectively. Gray: C; light blue: W.

## 3. Method

### 3.1. Calculation Details

The spin polarization density functional theory (DFT) calculation was performed by the *Vienna ab initio Simulation Package* (VASP) [32]. The projector augmented wave (PAW) basis was adopted, and the kinetic energy cut off was set to 500 eV. The generalized gradient approximation (GGA) methods in the scheme of Perdew–Burke–Ernzerhof (PBE) described the exchange and correlation potential [33]. The Brillouin region of bulk materials was sampled by a uniform k-point grid of $11 \times 11 \times 11$. The Brillouin region of $2 \times 2$ supercells was sampled by a $5 \times 5 \times 1$ uniform k-point grid. The convergence tolerances of the energy and force on each atom during structural relaxation were less than $10^{-6}$ eV and 0.01 eVÅ$^{-1}$, respectively. The vacuum layer was set at 15 Å. In this study, the $2 \times 2$ six-layer supercell plate model on the surface of WC (0001) is selected, such as reference [34]. The first two layers of WC (0001) surface with nonmetal atoms doped were relaxed, and the bottom four layers were fixed.

### 3.2. Theoretical Model and Doping Scheme

WC has a simple hexagonal structure [35], and its cell structure parameters are a = 0.2906 nm and c = 0.2837 nm [23]. By our calculations, the cell structure parameters were a = 0.29189 nm and c = 0.28446 nm, which were close to the literature values.

The WC (0001) surface is considered the most stable, and most research models are focused on the (0001) surface [33,36,37]. Therefore, the WC (0001) surface is selected as the research object in this study. The surface WC is composed of alternating W and C elements on the z axis. The surface of WC (0001) is a polar surface, which can be divided into W-terminal and C-terminal regions. In this work, we mainly study the substitution of the C element by nonmetallic element doping, so only the C-terminal model is explored.

Each C bond on the surface is connected to three W bonds on the subsurface, which makes a total of four C atoms with equivalent positions, numbered from 1 to 4, as shown in Figure 6a. At these positions, we substituted five different nonmetallic elements (B, N, O, P, and S) for C on the surface and varied the substitution concentration from 0.25 to 1.00.

### 3.3. Substitution Energy.

The substitution energies of the different nonmetallic doping elements can be defined as follows:

$$\Delta E_{sub(i)} = (E_{C_{1-n_i}X_{n_i}/WC(0001)} + iE_C) - (E_{WC(0001)} + iE_X) \tag{2}$$

where $X$ is the nonmetallic element (B, N, O, P and S), $E_{C_{1-n_i}X_{n_i}/WC(0001)}$ is the total energy of the WC surface with the nonmetallic element $X$ replacing the C atom, $E_{WC(0001)}$ is the total energy of the WC surface, $E_C$ and $E_X$ are the energies of one C atom and one nonmetal X atom, respectively, $i$ is the number of nonmetallic X atoms doped in the model, and $n_i$ is the concentration of doped nonmetallic elements.

*3.4. Gibbs Free Energy of Hydrogen Adsorption*

The Gibbs free energy of hydrogen adsorption ($\Delta G_H$) is defined as follows [38,39]:

$$\Delta G_H = \Delta E_H + \Delta E_{ZEP} - T\Delta S \tag{3}$$

where $\Delta E_H$, $\Delta E_{ZEP}$ and $\Delta S_H$, represent the energy difference of hydrogen adsorption, zero-point energy, and the entropy, respectively, between the adsorbed state and the corresponding freestanding state. $\Delta E_H$ is defined as:

$$\Delta E_H = E_{C_{1-n_i}X_{n_i}/WC(0001)+H} - \left( E_{C_{1-n_i}X_{n_i}/WC(0001)} + \frac{1}{2}E_{H_2} \right) \tag{4}$$

where $E_{C_{1-n_i}X_{n_i}/WC(0001)+H}$ is the total energy of adsorbing one H atom on the surface of WC(0001) after doping and the substitution of nonmetallic elements and $E_{H_2}$ is the energy of one $H_2$ molecule.

$\Delta E_{ZEP}$ can be determined by equation:

$$\Delta E_{ZEP} = E_{ZEP}^{H*} - E_{ZEP} - 1/2E_{ZEP}^{H2} \tag{5}$$

The $\Delta S_H$ can be approximated as

$$\Delta S_H \cong -\frac{1}{2}S_{H_2}^0 \tag{6}$$

where $S_{H_2}^0$ is the entropy of $H_2$ gas under the standard condition [39].

Due to the fact that the vibrational entropy in the adsorbed state is small according to the previous studies [38], Here, the values of $\Delta E_{ZEP}$ and $T\Delta S$ are referenced from ref. [40]. Therefore, $\Delta G_H$ can be written as

$$\Delta G_H = \Delta E_H + 0.3 \, eV \tag{7}$$

The optimal $\Delta G_H$ value for HER is close to 0 eV, which means that the smaller the $|\Delta G_H|$ value is, the better HER performance the catalyst has.

## 4. Conclusions

We investigated the doping of different various concentrations of nonmetallic elements into WC (0001) and their effect on the stability and catalytic properties of WC (0001). It is found that doping with nonmetal N and O can improve the surface stability, while the HER catalytic activity can be improved by doping with nonmetallic O and P doping. The catalytic activity of the surface, especially the Gibbs free energy of hydrogen adsorption ($\Delta G_H$) of P, is close to that of Pt. Charge transfer alone cannot be used to describe the change of $\Delta G_H$. Thus, we need to combine crystal structure and electronic structure changes to study HER activity and stability. In other words, the surface structure and electron changes caused by nonmetallic dopants co-regulate stability and HER catalytic activity. The electronic structure analysis shows that the p-d orbital hybridization caused by doping with nonmetallic elements leads to electron rearrangement near the Fermi level, which can improve the HER catalytic activity. This work provides a good theoretical basis for how to improve the catalytic activity of HER in the future.

**Author Contributions:** The study was planned and design by P.Q., X.X., X.W., DFT calculation were performed by Y.S., the manuscript prepared by Y.S., K.S., Y.Z., Q.L. All authors have read and agreed to the published version of the manuscript.

**Funding:** This research was funded by National Key Research and Development Program of China, grant number 2016YFB0700505 and 2018YFB0704300.

**Acknowledgments:** The authors gratefully acknowledge the financial support of National Key Research and Development Program of China (2016YFB0700505 and 2018YFB0704300).

**Conflicts of Interest:** The authors declare no conflict of interest.

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
