# Peer review of "Theoretical Study on Improving the Catalytic Activity of a Tungsten Carbide Surface for Hydrogen Evolution by Nonmetallic Doping"

_catalysts, doi:10.3390/catal10111272_

Round 1

Reviewer 1 Report

Having examined your manuscript entitled “Theoretical Study on Improving Catalytic Activity of  Tungsten Carbide Surface for Hydrogen Evolution by  Nonmetallic Doping I note that you have made some interesting measurements related to he effects of different types of non-metal (NM = B; N; O; P and S) and doping  concentrations on HER catalytic activity but before their publication several issues should be addressed. Therefore, it requires a mayor revision. Please check carefully.

  • Abstract, introduction and Conclusions

The author should highlight the relevance to develop NiO based nanomaterials.

The authors should contract their results with the detailed computational studies by Wannakao et alon a W-terminated WC(0001) surface, decorated with Mn, Fe, Co, Ni, Cu, Zn, Ru, Rh, Pd, Ag, Ir, Pt and Au monolayers

The authors should underline the innovative of the use of non-metal (NM = B; N; O; P and S), what is the advantage of these dopants selected. Please give examples and compare with the state of the art.

  • Materials and Methods

The authors selected the (0001) surface, What about another surfaces? You explore mainly the Pristine C-terminated, wat about the W-terminated WC(0001) surfaces.

  • Results and Discussion section:

How you can verify the sufficiency of this model, adsorption energies of a single dopant atom on the terminations of the carbide compared by using different layers of the support?

Have you used vibrational frequency analysis to verify the obtained transition state geometries

The effect of the support termination can also be established by analyzing the electron localization function of the system, have you studied that?

How the structure and electronic properties of the supported dopant are strongly influenced by the termination of the support,

The ttheoretical basis developed in this work  can work to improve the catalytic activity of HER of Cubic tungsten carbide a-WC as well as  hexagonal closed packedβ-WC ?

The authors discuss vaguely the applicability of this kind of compounds, and their requirements?

Therefore your article will not be considered, if the author does not include and add the changes proposed

Reviewer 2 Report

The authors studied the effect of doping on various properties of WC related to HER. The study could be interesting, but at the moment I feel that the language of the manuscript needs considerable improvement. I also feel that not everything is explained in detail, in a manner that can be easily digested by the readers. I suggest significant editing of the manuscript both of the content and of the language.

Page 3, how do we know that the ZPE should be 0.3 in equation 3? could this be explained?

page 3: „For the WC (0001) C-termination, the structural relaxation in the first two layers is 21.7%, while the second and third layers is 6.23%. 120” I may misunderstand the situation, but to the authors refer to the distance between the layers or to the layers themselves? This should be more clearly explained.

line 124: put the reference number! (those of Gaston N et al.)

line 103: “is the energy of one H molecule. 104 E” do they talk about absorbing a H2 molecule or a H atom? I think the expression H molecule is very unprecise and should be avoided. Similarly in the figure caption of Figure 2 it is not clear what is absorbed on the surface. I think the manuscript could be improved by adding a figure of the proccess that they model.

line 142: „As shown in Table 4, the most stable 142 adsorption site distribution of a H atom adsorbed on different surface doping elements and 143 concentrations, it can be seen that the adsorption sites of the non-metal N, O, P and S adsorption 144 energies are all top.” I don’t fully understand the sentence and I think they want to refer to Table 2 and not to Table 4.

Between lines 150-158 the authors describe the trends they observed for the absorption energy. Could they give any explanation why B doping is so favoruable? how does the geometry of the doped surfaces look like, and how this changes with dopant and dopant concentration.

I don’t understand this: “For the other four unstable nonmetallic elements, the ΔEsub(i) of N is relatively small, and is relatively stable. 136” what is N?

I wonder how spin is treated in the calculations? Could the author elaborate a little bit on this in the manuscript?

Figure 3. The authors could explain a little bit better which charge they mean, and also explain why the charge is highest on N and O doped systems (I am sure the calculated numbers are fine, but it is either surprising or not enough explanation is given to be able to fully understand the trends)

some examples of sentences, expressions that seem to be incorrect:

(I only collected some from the first pages of the manuscript)

page 1 the enough reaction rate.

page 1  and higher  resistant poison.

page 2” transition metal (Co, Ni and Fe) doped WC could improved its 54 catalytic activity.”

could effectively enhanced

„However, especially, there hardly has works systemically study the non-metallic 57 element doping in WC.”

„In this study, the 2x2x1 six-layer supercell plate model on the surface of WC (0001) 67 is selected to discuss [21].”

„The generalized gradient approximation (GGA) methods in the scheme of Perdew–Burke–Ernzerhof (PBE) to describe the exchange and correlation potential” (there is no verb in the sentence)

Reviewer 3 Report

The manuscript reports a theoretical study of the hydrogen evolution reaction (HER) on C-terminated tungsten carbide (0001) catalysts using density functional theory.
The author's aim is to investigate the effect on hydrogen atom adsorption, if carbon in the topmost surface layer is successively replaced by the elements B,N,O,S.
As also pointed out in the introduction section, the improvement of the catalytic activity of tungsten carbide is a topic of high interest in catalysis.
Therefore, the manuscript would attract a broad readership in 'Catalysts'. However, the manuscript needs considerable improvements before I can
recommend its publication.

General design of the study:
If I have understood correctly, the carbon atoms in the surface layer are successively replaced by non-metallic atoms at sites 1,2,3, and 4, as shown in Fig. 1a. So, going from 75% to 100% doping density, carbon at site 4 is replaced. For hydrogen adsorbed on top of site 4 (point f shown in Fig. 1b), the atom species underneath changes from carbon to N, O, P. It is thus obvious that e. g. the large change in adsorption energy from -1.98577 eV to +0.25959 eV results from the change of the chemical identity of the atom directly underneath the adsorbed hydrogen. Before discussing changes of the density of states near the Fermi level, charge-transfer etc., this (trivial?) aspect should be mentioned in the first place. But I wonder if this should be called an effect of (surface-)doping if the complete terminating carbon layer is replaced e. g. by oxygen.

Equation (3) (line 108):
The authors provide a formula for the change of the Gibbs free energy of adsorption of atomic hydrogen, neglecting the change of entropy and by setting the change of zero-point energy (ΔEzep) ad hoc to apparently 0.3 eV (unit eV is missing in the formula!). For this, the authors cite reference 25, but in this article, this value for the zero-point energy correction is not explicitly mentioned. The authors should provide more information on this value, since it is critical for the interpretation of the further results presented in table 2 and Fig. 3a. Moreover, in equation (3) the symbol 'Δ' is missing before GH, and in line 103 'one H molecule' should read 'one H2 molecule'. Such inaccuracies do not strengthen the reader's confidence in the presented data at all.

Line 195, Fig. 4 : The authors state, that the DOS pattern of WC(0001)P4 is 'most similar to Pt'. Since the authors do not provide any DOS profiles of Pt, the reader cannot assess this statement.

Line 179 and line 205: The authors state in line 179, that it is 'not appropriate to use the electron transfer trend to explain the change in  ΔGH'. In contrast, in the conclusion section, line 205, they conclude that 'electron transfer is a fine descriptor to analyze the relationship between ΔEsub(i) and ΔEH'. Can the authors clarify to what extent these statements are compatible with each other?

Style and grammar:
The whole manuscript needs revision with respect to style and grammar. Just to mention one example: Sentences like 'However, especially, there hardly has works systematically study the non-metallic element doping in WC.' (line 57) are almost incomprehensible to the reader.

References section:
Behind the title of each reference, there appears a redundant '[J]' in the manuscript. In reference 34 (line 291), the first name of the first author is misspelled: 'N?Rskov' should read 'Nørskov'.

Round 2

Reviewer 2 Report

The authors have improved the manuscript considerably. The language style is better and results are explained more understandably.

A few minor remarks:

(1)Page 3: "where∆?H, ????? , ?and ∆?H, represent different hydrogen adsorption energy, different
110 zero-point energy, temperature and the different entropy between atomic hydrogen adsorption and
111 hydrogen in the gas phase, respectively. "

What does different hydrogen absoprtion energy mean? The word different is used very strangly in this sentence. 

(2)For the WC (0001) C141 termination, the structural relaxation between the first two layers is 21.7%, while the distance 142 between the second and third layers is 6.23%

It is strange to read "the distance 142 between the second and third layers is 6.23%". How can a distance be 6.23%? 

(3) ELF is electron localisation function and not electron local function as said on page 8. 

Author Response

We have carefully reviewed the reviewer’ reports. We have addressed each and all questions that the reviewers may have in the revised manuscript. The changes made in the text are highlighted for your examination.

   The main revisions we have made are listed as following:

  1. According to the reviewer #2 suggestion, we have updated the description on page 3, line 110-111.
  2. According to the reviewer #2 suggestion, we have improved the description for the change of interlayer distance after structural relaxation on page 4, line 141-142.
  3. According to the reviewer #2 suggestion, we have changed the “electron local function” to “electron localisation function” on page 8, line 231-232.

Reviewer 3 Report

After consideration of the new version of the manuscript titled "Theoretical study on improving catalytic activity of tungsten carbide surface for hydrogen evolution by nonmetallic doping" by Ye Su et al., I come to  the conclusion that the manuscript is now acceptable for publication in 'Catalysts'.

Author Response

Thanks for your agreement  to publish our manuscript in  'Catalysts'.